# Domain Word Extension Using Curriculum Learning

**DOI:** 10.3390/s23063064

**Published:** 2023-03-13

**Authors:** Sujin Seong, Jeongwon Cha

**Affiliations:** 1Smart Environmental Engineering, Changwon National University, Changwon 51140, Republic of Korea; 2Computer Engineering, Changwon National University, Changwon 51140, Republic of Korea

**Keywords:** token expansion, curriculum learning, pre-trained models

## Abstract

Self-supervised learning models, such as BERT, have improved the performance of various tasks in natural language processing. Although the effect is reduced in the out-of-domain field and not the the trained domain thus representing a limitation, it is difficult to train a new language model for a specific domain since it is both time-consuming and requires large amounts of data. We propose a method to quickly and effectively apply the pre-trained language models trained in the general domain to a specific domain’s vocabulary without re-training. An extended vocabulary list is obtained by extracting a meaningful wordpiece from the training data of the downstream task. We introduce curriculum learning, training the models with two successive updates, to adapt the embedding value of the new vocabulary. It is convenient to apply because all training of the models for downstream tasks are performed in one run. To confirm the effectiveness of the proposed method, we conducted experiments on AIDA-SC, AIDA-FC, and KLUE-TC, which are Korean classification tasks, and subsequently achieved stable performance improvement.

## 1. Introduction

After training to understand the characteristics of the language for large-scale corpora, the pre-trained language models are applied to downstream tasks in various natural language processing fields, showing excellent performance. However, since most of these models are trained in a general domain, the characteristics of the pre-trained language models are biased to general vocabulary. If there is an abundance of domain-specific vocabulary in the downstream domain data, it is even more difficult to the fine-tuned language models. Since the existing pre-trained language models do not have sufficient information about a specific domain, we should show the meaning of domain-specific words through the context embedding of the sub-word. Recent works have shown that using an optimized vocabulary for a specific downstream domain is effective, and thus shows the advantage of having a fixed embedding. This led to a study on adding vocabulary with a domain-specific word in the pre-trained language models [1,2,3].

The previous main approach was to learn the pre-trained model from the beginning with specific-domain data or to conduct an additional second pre-trained training using the existing pre-trained model using domain training data [4,5,6,7,8,9]. These methods not only require a large amount of time for computation but also require large-scale data for each domain for training. Recently, research has been conducted to expand the vocabulary of language models without an additional domain corpus and to learn appropriate training about new vocabulary. Therefore, these studies adopt an extension module to the existing pre-trained language models structure to help adapt to extended vocabulary embedding.

We propose a learning method that provides a new embedding value that is improved without changing the system structure of the pre-trained language models. We first conduct the initial learning with data from the downstream task. In this case, we do not add a new token. In other words, the pre-trained model is adapted to the downstream task. We call this step the warm-up stage. The next step is to add a new token and re-tuning. This step makes tokens, which did not learn, that will be learned according to the downstream task. Figure 1 shows this process.

The contributions of the proposed method are summarized as follows:We suggest adding new tokens for domains without learning the pre-trained model.We conduct experiments with various data to show the performance of the newly built model.

## 2. Related Work

BERT [10] is a popular pre-trained language model used for NLP research specialized in classification tasks built using English Wikipedia, BookCorpus [11]. However, we need to identify the best performance for a non-trained domain such as life science.

The most intuitive way to solve the problems caused by domain disagreements is to build separate language models for the domain.

Scibert [4] creates language models from the data of computer science and broad biomedical domain randomly sampled from Semantic Scholar [12]. PubMedBERT [5] made the domain range narrower than SciBERT, using PubMed data to configure the vocabulary of a language model for the biomedical domain and perform pre-training of the model from scratch. These models demonstrate better performance than the general domain language models. However, in order to learn the language models, it is expensive to apply to each domain because it requires a large corpus and a long learning time.

Another approach is to continue pre-training using target domain data based on the existing pre-trained language models. BioBERT [7] initialized the model with the weight of the language models pre-trained in the general domain and was second pre-training with the PubMed abstracts and PMC full-text Articles. Continual pre-training can transfer the weight optimized for the general domain to the target domain, which can take relatively less training time to improve the fine-tuning performance. However, the vocabulary used is still the same as that of the original model created based on a general domain. It also still requires a large amount of fine-tuning data.

It has been proven helpful to improve performance by including domain-specific vocabularies in language models used to perform tasks for specific domains. In recent studies, the domain-specific vocabulary is added to the existing pre-trained language models, and the extended embedding is undergoing continual improvement so that the weight of the existing language models can properly handle it. exBERT [1] uses a merged result by adding the extension module to the existing BERT to adapt to the new vocabulary after adding the wordpiece-based domain-specific vocabulary. In the domain pre-training, only the new embedding and the weight of the expansion module are learned, and in fine-tuning, only some of the top layers are learned. This learning method can reduce the computation in pre-training and fine-tuning; however, modifying the existing pre-trained language models structure is necessary.

Avocado [2] extracts extension vocabulary that reflects the relative importance of words using only downstream task datasets and normalizes the embedding of the expansion vocabulary in the fine-tuning process. For the normalization of the extended vocabulary embedding, an additional model must perform the prediction through the input separated by the existing tokenizer, so there is twice as much memory as the vanilla fine-tuning.

Sachidananda et al. [3] extracted the expansion vocabulary based on the difference between the characteristics of the tokens in the downstream task dataset and the characteristics of the tokens in the existing pre-trained language model domain datasets. The subword embedding and the project-based initialization average are considered for initializing the embedding value of new vocabulary. They confirmed that using the average value of the subword increased speed two-fold and demonstrated similar performance.

From previous studies, we confirmed that adding a domain-specific vocabulary and initializing them to the average of subword embedding helped improve performance. We want to expand the vocabulary and improve the method of initializing the average of subword embedding.

## 3. The Proposed Method

The pre-trained language models have a fixed vocabulary and weight the models. The vocabulary that the pre-trained language models can hold is bound to have a limited number based on general-purpose data for the model’s generality. However, unlike general written and spoken language, many vocabularies are used only in medicine and law. It is difficult to express specialized domains properly using a vocabulary limited to the general domain. We suggest an effective way to add vocabulary important in the downstream task domain but not in the pre-trained language models.

### 3.1. Extracting Domain Specific Vocabularies

The domain-specific expansion vocabulary is a vocabulary that cannot be expressed sufficiently due to the limitations of the vocabulary of the pre-trained language models. In other words, these words are more frequent in documents in certain domains than in general domain documents, and the frequency should be larger enough to impact training significantly. In addition, the expansion vocabulary is not duplicated with the vocabulary of the pre-trained language models.

We utilize training data from downstream tasks to automatically extract the most suitable extended vocabulary for the task based on a fixed number. Initially, the vocabulary list is populated with all characters in the dataset. Next, We identify character pairs with the highest probability (TokenScore) of occurring sequentially and merge them to create a new token. This token is then added to the vocabulary list.
$$\mathrm{Token}\;\mathrm{Score}=\frac{FREQ(c_{i}c_{j})}{FREQ\left(c_{i}\right)\times FREQ\left(c_{j}\right)}$$

To illustrate, if “base” and “line” are the most frequently occurring terms in “base + line”, they will be merged first and the new token “baseline” will be added to the vocabulary list. Conversely, if “base” and “line” appear frequently but separately in other words such as “base + ball” and “line + number”, they will be merged later in the process. This iterative process continues until the number of tokens in the vocabulary list reaches a predetermined maximum, at which point the vocabulary list update is terminated. This method is similar to the approach used by existing pre-trained language models such as BERT and Electra’s Wordpiece Tokenizer training method, as the extended vocabulary is organized in the same format as the vocabulary list of the pre-trained language models. This makes it easy to add new vocabulary to the list. Using the automatically extracted and added extended vocabulary lists in downstream tasks, the tokens that were previously decomposed into UNK tokens by pre-trained language models can now be used to store their meanings. Additionally, overly tokenized words can now be tokenized into significant units if the merged token is added as an extended vocabulary.

Figure 2 displays the tokenization results obtained from pre-trained language models based on Korean newspapers and Wikipedia documents. As the tokenizer is trained on Korean data, it may have limited effectiveness in handling other languages. The sentence “$X^{2}$ test with titanic datasets” in Figure 2 is tokenized to [“[UNK]”, “test”, “with”, “titanic”, “d”, “ata”, “se”, “ts”]. In the science and technology domain, “$X^{2}$” means the square of standard normal distribution but loses its specific meaning since it is tokenized with the unknown token. The "datasets" are a collection of various data stored in digital format, often used in scientific and technical documents containing experiments. However, “datasets” is split as [“d”, “ata”, “se”, “ts”], making it difficult to directly represent the meaning of “data” and “set”.

We can make a meaning for the vocabularies that are useful for downstream tasks using a tokenizer with extended vocabularies.

Figure 3 illustrates the impact of adding extended vocabulary extracted through the tokenizer, learned from scientific paper data, to the tokenizer models shown in Figure 2. The downstream data-based tokenizer can extract $X^{2}$ as an extended vocabulary, even though it was not present when the pre-trained language models were trained. As a result, the general-domain tokenizer can identify the previously UNK tokenized $X^{2}$ as a “$X^{2}$” token, which may have a different value than other UNK tokens. Additionally, the word “datasets” can be decomposed into “data”, “set”, and “s”—significant components in the downstream task data. By using the token “data” instead of the individual letters “d” and “ata”, we can assign a specific meaning that is more relevant to the context. The letter “d” appears in so many different types of words that it is difficult to provide a specific meaning, while “ata” is usually used in plant names such as “crenata” and “moschata” in scientific data. However, by adding the token “data” to it, we can represent the meaning of “information used to train the models for downstream tasks”. Adding the token “dataset” as a single term may be more effective than dividing it into “data” and “set”. However, it was not included as an extended vocabulary due to variations in the term used in Korean science and technology papers used to train the extended vocabulary tokenizer. The term appeared in various forms such as “data” and “data set”, making it difficult to standardize. Therefore, it means the addition of “dataset” as an extended vocabulary has less impact than adding “data” and “set” in downstream tasks.

### 3.2. Creating the Extended Word Embedding and Application Method

The extended vocabulary specialized in the domain presents a new vocabulary that the existing pre-trained models have never trained, so it is important to allow the pre-trained models to handle the new vocabulary properly. If new word embeddings are not properly processed by the models and behave as noise, we assume that they do not properly reflect the meaning of the vocabulary. In that case, this can reduce model performance because the amount of information about the sentence given by input is reduced. In response, we propose a training method that allows the pre-trained models to handle the new word embedding with the old word embedding.

Sachidananda et al. [3] have shown that initializing new expansion word embeddings as the average of their subword embeddings is more efficient than training a new embedding for the expansion vocabulary with an additional models and using it as the initial value. Although the sub-words in the expansion vocabulary may not be meaningful on their own, they still have the advantage of remaining within the distribution of embeddings that the existing pre-trained language models have.

However, all the subwords of the expansion vocabulary are less related to the meaning in a specific domain and are not sufficiently trained data containing the vocabulary. It is difficult to change the existing meaning of the pre-trained models to the meaning used in the downstream task domain.

For example, ‘baseline’ means ‘model or experimental result that is the standard for performance comparison’ in the science and technology domain. However, when we separate this word using a general domain tokenizer, we usually get ‘base’ and ‘line’. ‘Base’ means the lowest range or baseball base. ‘Baseline’ means the lowest range in the music field and the line between the baseball base in the sports field. Therefore, there is a difference in the meaning used in science and technology. The new vocabulary of ‘baseline’ is still in the meaning of the general domain by using the average meaning of ‘base’ and ‘line’, so it required more than the data size already trained to transfer the meaning of the science and technology field.

We introduce curriculum learning to provide better initial values for new expansion vocabulary without transforming or adding the structure of the existing pre-trained language models. The curriculum learning [13] is a learning strategy formulated in the context of machine learning that humans and animals learn better when training in the order of from easy to complex examples. At first, you train the models with an “easy sample” to identify the outline of the whole and gradually learn complex examples to find the best results. Curriculum learning makes a faster convergence of the learning process and guides the learning machine so that the initial learning step can be better transferred to the final learning step. From the point of view of curriculum learning, the newly added expanded vocabulary corresponds to a “difficult” sample where the weight of the existing pre-trained language models have not been learned. On the contrary, the vocabulary of the existing pre-trained language models are an “easy” sample that has already learned enough meaning. We train both the embedding and weight of the existing pre-trained language models about the domain data and transfer the meaning of the vocabulary to the target domain. Then, using the subword’s embedding, we initialize the extended vocabulary embedding and perform fine-tuning.

Since subword embeddings and models weights are re-trained in Downstream Task Domain, the average of subword embedding is also not a big difference in models weight and distribution. In addition, since each subword embedding has been learned with domain data, the initial value of the new expansion vocabulary embedding can be set close to the value used in the corresponding domain. Figure 4 illustrates the structure of the models that perform the proposed method.

Algorithm 1 outlines the process for setting up an embedding of a domain vocabulary $V_{D}$, which consists of a base vocabulary $V_{B}$ with an extended vocabulary $V_{A}$ added. The model s*M* that predict output *y* for input *x* have a weight parameter θ, which the embedding module *E* uses to map input tokens *V* to embedding vectors. The first step is to conduct a warm-up training phase based on a pre-trained language models. During this phase, the input sentence is tokenized by a tokenizer with $V_{B}$ ($tok(x,V_{B})$), and a pre-trained language models with a randomly initialized classification layer is trained to predict the classification label *Y*. This phase is similar to the fine-tuning phase of transfer learning with a pre-trained language models. The warm-up phase ends when the models are early-stopped based on the performance of the validation data measured during training. During the warm-up phase, the models $M(\hat{\theta })$ optimize the randomly initialized classification layer for the downstream task and translates the existing token embeddings into the meaning space of the downstream task domain. As a result, the token embeddings in the language models become more closely aligned with the meaning of the downstream task domain than the initial embeddings. In other words, the warm-up phase allows the language models to adapt to the specific vocabulary and linguistic patterns of the downstream task, leading to improved performance. Next, the models $M(\hat{\theta })$ are fine-tuned with downstream task data by adding an extended vocabulary $V_{A}$. To extract $V_{A}$, a separate tokenizer is trained on the training data of the downstream task, and all tokens are added except for those that are single-letter or that already exist in the base vocabulary $V_{B}$ of the existing pre-trained language models. By adding $V_{A}$ to $V_{B}$, the tokenizer obtains an updated vocabulary list $V_{D}$ that includes the extended vocabulary. The tokenizer then uses $V_{D}$ to tokenize the downstream task data, and the models are fine-tuned on this tokenized data to adapt to the specific vocabulary and linguistic patterns of the downstream task. Since the embeddings for the added extended vocabulary are not in the existing pre-trained language models, we add them by initializing them with the average embedding of the sub-words $E(v,\hat{\theta })$ (Algorithm 1 lines 5–10). The models $M(\hat{\theta })$ trained for the downstream task is retrained to predict the paired labels *Y* given a sentence $tok(x,V_{D})$ tokenized by a tokenizer with $V_{D}$ as input. The models are trained until it is early-stopped based on performance on the validation data, adapting the embeddings for the extended vocabulary to the downstream task. After fine-tuning, the terminated models are used to make predictions on new data.
**Algorithm 1** Warm-up Training Adapted Models**Require:** Vocabulary of Base, Extended, and Domain $V_{B},V_{A},V_{D}$, Tokenizer Tok, Input Sentence *X*, Ground Truth *Y*, Base LM M, Embeddings Module *E*1:# Warm-up Training M:X→Y with SGD using Base Vocabulary $V_{B}$2:$\hat{\theta }\leftarrow \underset{\theta }{\mathrm{argmin}}\left(-1\right)\times \sum Y\log_{e}M\left(Tok\left(X,V_{B}\right),\theta \right)$3:# Update Embedding *E* for extended vocabulary $V_{A}$4:$V_{D}\leftarrow V_{B}+V_{A}$5:**for** *v* in $V_{D}$ **do**6:    **if** *v* not in $V_{B}$ **then**7:        $S_{A}\leftarrow Tok\left(v,V_{B}\right)$               ▹$S_{A}$ is subwords of vocab *v*8:        $E\left(v,\hat{\theta }\right)\leftarrow \frac{1}{|S_{A}|}\sum E\left(S_{A},\hat{\theta }\right)$9:    **end if**10:**end for**11:# Fine-Tuning M:X→Y with SGD using Domain Vocabulary $V_{D}$12:$\theta^{*}\leftarrow \underset{\hat{\theta }}{\mathrm{argmin}}\left(-1\right)\times \sum Y\log_{e}M\left(Tok\left(X,V_{D}\right),\hat{\theta }\right)$13:**return**$\;\theta^{*}$

## 4. Results

### 4.1. Data

We use three datasets as follows:

AIDA-SC: The Korean thesis meaning tagging dataset, which was released in AIDA (https://aida.kisti.re.kr (accessed on 30 December 2022)), has a label that indicates the intention of the sentence in the Korean science paper. In the sentence, nine labels indicate the intention of the sentence, including ‘hypothesis setting’, ‘technology definition’, ‘target data’, ‘data processing’, ‘problem definition’, ‘performance/effect’, and ‘theory/models’.

AIDA-FC: “Research topic classification data” released in AIDA aims to provide the title of Korean papers, journal names, and full papers and to create science and technology standard classification codes for them. The science and technology classification code has first-level and second-level codes, and we only use the first-level codes.

KLUE-TC: The KLUE (https://github.com/KLUE-benchmark/KLUE (accessed on 30 December 2022)) topic classification tasks unveiled in KLUE aim to classify appropriate news categories for the news title given. Since the evaluation data are not disclosed, we used the verification data to evaluate the performance of the models, and 10% of the training data were used as the verification data.

Table 1 summarizes the information of each downStream rask and dataset.

### 4.2. Pre-Trained Models for Classification

We choose KLUE-BERT-base [14], KoElectra [15], KorSciBERT and KorSciElctra (https://aida.kisti.re.kr (accessed on 30 December 2022)) as pre-trained language models to add extension vocabulary modules. We deliver the output of the final layer for the first input token to the linear layer to predict. Table 2 shows the number of extended vocabularies added per pre-trained language models for each downstream task. Among the pre-trained language models, KorSciBERT uses a part-of-speech tagger for preprocessing to first separate words into their smallest semantic units before tokenizing them using wordpiece tokenizer. As a result, tokens larger than the size of the words separated by preprocessing cannot be applied in the tokenization process. Therefore, the number of extended vocabularies added to KorSciBERT is relatively small compared to the other pre-trained language models.

### 4.3. Experimental Results

We compare the vocabulary of the pre-trained models and the extension vocabulary that adds the domain-specific vocabulary. We also compare the curriculum learning method with a simple average initialization method for setting the initial value of the extension vocabulary. Table 3 shows the experiment’s performance, and all performance was measured by Macro F1-Score.

### 4.4. Discussion

As can be seen in Figure 5, a high-performance improvement rate is observed when the domain of data used in the pre-trained language models and the domain used in the domain of downstream tasks are different. When performing the task of the different domain, we can obtain improved performance by adding only the extension vocabulary initialized as an embedding average in the pre-trained model without any additional work; however, it is generally slightly lower when performing the same domain task. You can see that where we have applied curriculum learning, we generally achieve improved performance across all domain tasks.

If you add an expansion vocabulary, the word that was tokenized to a small length can be token to be longer. Long tokens can eventually transform many letters into one token, so the number of tokens in the sentence can be reduced. Since there is a limit to the input length of the language models, reducing the number of tokens in one sentence means that the input can deliver more words of the model. Figure 6 shows the increase and decrease in F1-score according to the rate of reducing the longest sentence length in data. You can see that the performance improves greatly when the tokenized sentence length decreases.

The new tokenizer splits the words in the downstream task into more meaningful sub-words than the tokenizer using the vocabulary of the pre-trained language models. To analyze the effects of the expansion vocabulary, we show the sample words of each domain token with the expansion. In order to analyze the effects of the expansion vocabulary, we show the sample word of the domain in Table 4 that was tokenized by the vocabulary of the pre-trained language models and expansion vocabulary applied to the KLUE-BERT-base model.

Since AIDA-FC is targeted at the paper’s title, there is a lot of expansion vocabulary for words with nouns, noun transformative endings, and noun derivative suffixes. AIDA-SC had many extended vocabularies on analytical methods that were not well represented in general domains, such as ANOVA and SPSS. In addition, unlike AIDA-FC, it has a full paper, so there is a lot of frequency of expansion vocabulary that uses nouns and particles as a vocabulary together. Since KLUE-TC targets the title of the news article, there is a lot of expansion vocabulary about the company name and person’s name, which is clearly distinguished from the AIDA datasets of the science and technology domain.

For example, one of Korea’s credit card companies, ‘우리카드 (/Uri kadeu/, Wooricard),’ is decomposed into ‘우리 (/Uri/, we)’ and ‘카드 (/Kadeu/, card)’ in the existing pre-trained language models. When we searched for a vocabulary category in large Korean news dataset (2021 version of everyone’s corpora), the word ‘우리 (us)’ appeared in the society field, and ‘카드 (/Kadeu/, card)’ appeared in the economy field. ‘우리카드 (/Uri kadeu/, WooriCard)’ has appeared a lot in the sports field. In the downstream task, ‘WooriCard’ is a lot in the ‘sports’ field. When we looked at the occurrence ratio of the vocabulary, when ‘우리 (/Uri/, we)’ and ‘카드 (/Kadeu/, card)’ appeared in the general corpus, the vocabulary corresponding to ‘우리카드’ was 53.97%, but occurrence rate in the downstream task data is 95.96%. The vocabulary of ‘우리카드’ is likely to have a higher importance in the downstream task domain.

This change in ratio is larger in the extended vocabulary of AIDA-FC, whose domain is significantly different from that of the general corpus. For example, ‘페이스트 (/Peist/, paste)’, a word that represents puree and ketchup, is decomposed into ‘페이스 (/Peis/)’ and ‘트 (/Teu/)’ in the existing pre-trained language models. In a general domain, ‘페이스 (/Peiseu/)’ is used in the sense of face and speed, and many appear in articles in society field. In the case of ‘트 (/Teu/)’, it is a letter that does not have a special meaning individually, and it appears in articles in society field and life areas. Relatively, in downstream tasks, ‘페이스 (/Peiseu/)’ is often seen at a similar rate in cultural arts and information and communication field, and ‘트 (/Teu/)’ is the most common in the ’information and communication’ field. In the case of ‘페이스트 (/Peiseut/)’, ‘페이스트 (/Peiseut/)’ is often found in the ’material’ field. When ‘페이스 (/Peiseu/)’ and ‘트 (/Teu/)’ appear simultaneously, ‘페이스 (/Peis/)’ appear only 2.34% in the general domain corpus, but appears 50% in the title of the science and technology filed corpus.

If we add expansion vocabulary, the frequency of UNK tokens in the tokenized data decreases. In the case of UNK tokens, it does not reflect the meaning of individual tokens and causes clear meaning loss because several vocabularies are expressed only as one vector. It can be separated using the added extended vocabulary, which can help improve performance. AIDA-SC and AIDA-FC are the same science and technology domain but show that the types of vocabulary used in the text of the paper are distinguished. In particular, many symbols are used in vocabulary or formulas that represent measurement units, such as μg and μm, which are not used in the extension’s title added to the AIDA-SC.

## 5. Conclusions

We experimented with Korean classification data based on curriculum-learning-based training methods to add domain vocabulary by minimizing code modification and additional operations to the existing pre-trained language models. First, we learn the models with tokenized inputs to the pre-trained tokenizer. We then change to an expanded tokenizer to conduct continuous learning. Through this, the embedding of the extended vocabulary, which is initialized as the mean of subword embedding, can attain a value closer to the meaning of the domain. Aside from an update of the tokenizer and embedding matrix in the middle of training, there is no modification of the model structure and is easily applied to the existing method because it only learns the optimization of the sentence that reflects the updated vocabulary. The proposed method shows that the performance improvement is greater than the model initialized as an embedding of the existing pre-trained language models. In particular, the performance is greatly improved when there exists a large domain difference between the pre-trained language models and the downstream task. As a result, the proposed method can effectively expand the vocabulary of the pre-trained language models through small modifications. In the future, we plan to increase the efficiency of the extended vocabulary added by adjusting the number of tokens added.

## Figures and Tables

**Figure 1 sensors-23-03064-f001:**
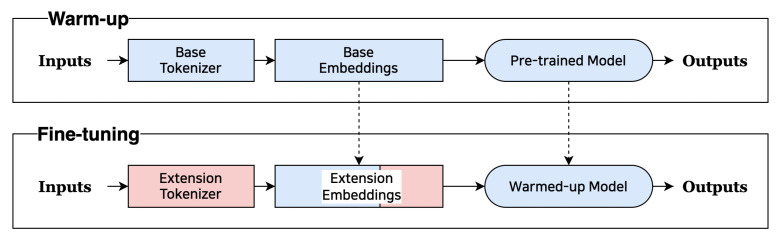
Concept of the entire process. The blue color displays the warm-up step, and the red color shows the expansion token and the fine tuning. The solid line shows the training progress, and the dotted line represents the use of embeddings and models.

**Figure 2 sensors-23-03064-f002:**
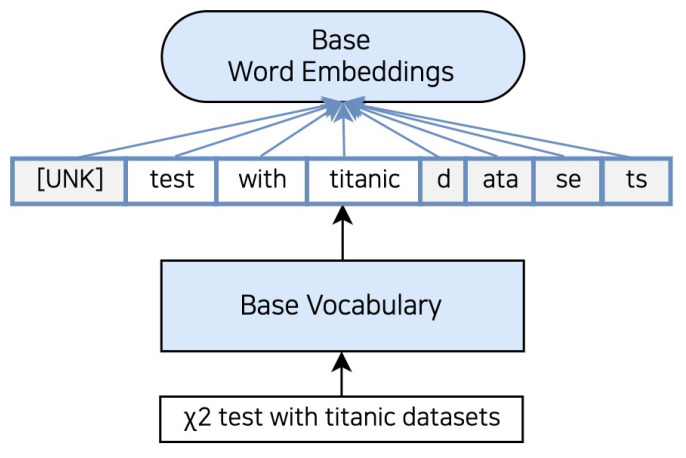
Examples using the tokenizer of the pre-trained language models. The input sentence is tokenized as vocabulary learned in the pre-trained and uses the pre-trained language models embeddings. For example, when you tokenize the “$X^{2}$ test with titanic datasets” using a base tokenizer, $x^{2}$ is a token to the “[UNK]” token. In addition, “Datasets” is divided into “D”, “ata”, “se”, “ts”.

**Figure 3 sensors-23-03064-f003:**
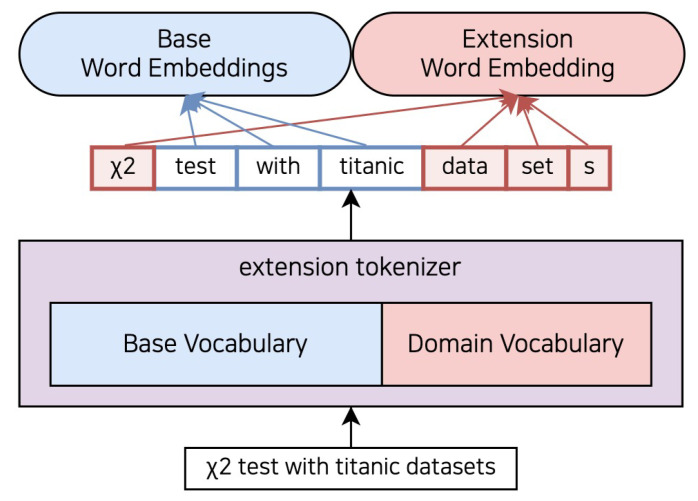
Examples using a tokenizer with a domain-specific vocabulary. The input sentence is tokenized as vocabulary, and domain-specific vocabulary is learned in the pre-training. The extended vocabulary uses the extended embedding added to the existing pre-trained language models embeddings. For example, in the “$X^{2}$ test with titanic datasets” sentence, $x^{2}$, a token as ’[UNK]’ through the existing pre-trained tokenizer, is a token to be ‘$x^{2}$’. In addition, “Datasets” can be divided into ‘data’, ‘set’, and ‘s’.

**Figure 4 sensors-23-03064-f004:**
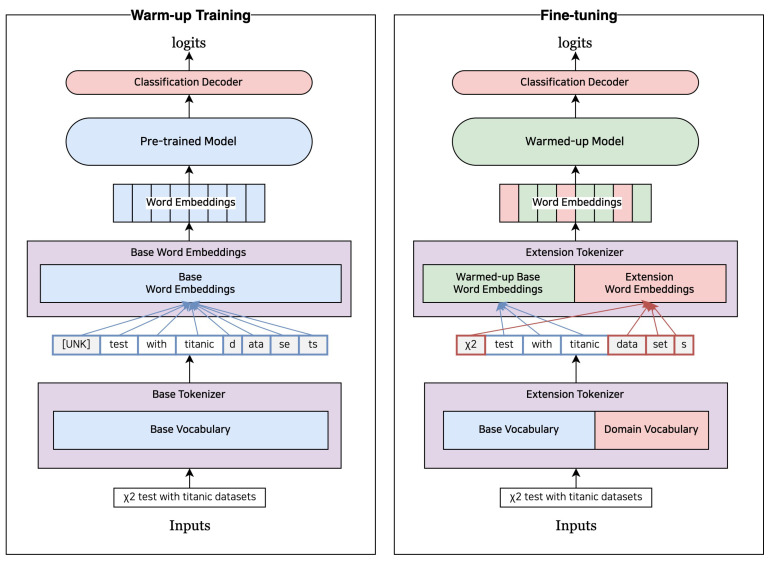
Model architecture using curriculum learning. Fine-tuning is performed in two stages by introducing a warm-up. Blue: vocabulary or model weights about base pre-trained models, Red: extension vocabulary or word embeddings, Green: updated model weights or word embeddings after warm-up.

**Figure 5 sensors-23-03064-f005:**
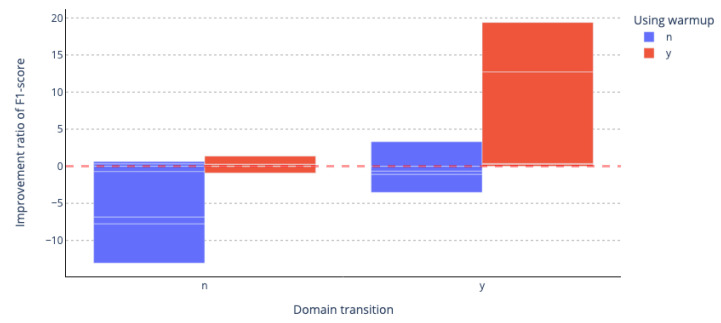
F1-score’s increase and decrease rate according to the presence or absence of prior learning and downstream domain differences. On the x-axis, n is a domain, and the same applies for the y-axis. Blue bars: experiments that do not use warm-up; red bars: experiments using warm-up. Pink dotted line: baseline performance.

**Figure 6 sensors-23-03064-f006:**
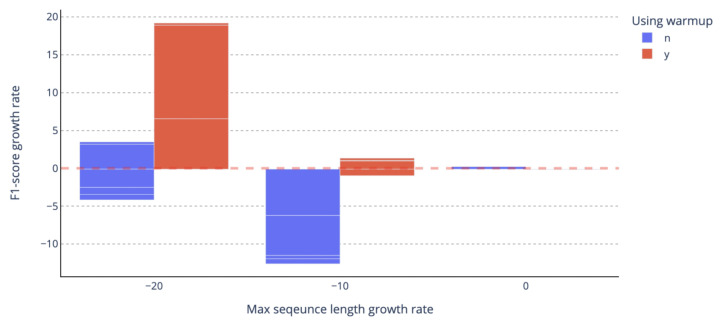
F1-score’s increase and decrease rate according to maximum token length decrease rate. −20 means an experiment in which the maximum token length decreased by more than 10% and less than 20%, and −10 means an experiment in which the maximum token length decreased by more than 0%, and less than 10% represents summation. Pink dotted line: the value when there is no change in performance.

**Table 1 sensors-23-03064-t001:** Specifications of the downstream task and datasets. Each number of data is the number of sentences.

Domain	Task	Train	Dev.	Test	# of Classes
Science & Tech	AIDA-SC	124,592	15,574	15,574	9
Science & Tech	AIDA-FC	24,000	3000	3000	33
News	KLUE-TC	41,110	4568	9107	7

**Table 2 sensors-23-03064-t002:** The number of extended vocabularies. Extended vocabularies added to the *A* and WA models are the same. *A* is a model with extension vocabulary. WA is a model with extension vocabulary and curriculum learning.

Pre-Trained Model	AIDA-SC	AIDA-FC	KLUE-TC
KLUE-BERT${}_{A}{/}_{WA}$	8854	10,477	7703
KoElectra${}_{A}{/}_{WA}$	8804	10,414	7684
KorSciBERT${}_{A}{/}_{WA}$	2310	5981	-
KorSciElectra${}_{A}{/}_{WA}$	8126	9434	-

**Table 3 sensors-23-03064-t003:** Result of different pre-trained models with or without extended vocabulary. *A* is a model with extension vocabulary. WA is a model with extension vocabulary and curriculum learning.

Domain	Model	AIDA-SC	AIDA-FC	KLUE-TC
General	KLUE-BERT	89.92	39.71	86.01
	KLUE-BERT${}_{A}$	89.52 (−0.40)	41.02 (+1.31)	85.39 (−0.62)
	KLUE-BERT${}_{WA}$	90.00 (+0.08)	42.36 (+2.65)	86.02 (+0.01)
General	KoElectra	89.50	29.78	84.33
	KoElectra${}_{A}$	88.91 (−0.59)	29.06 (−0.72)	84.59 (+0.26)
	KoElectra${}_{WA}$	89.75 (+0.25)	33.46 (+3.68)	84.56 (+0.23)
Science and Tech	KorSciBert	89.82	42.91	-
	KorSciBert${}_{A}$	90.11 (+0.29)	40.28 (−2.63)	-
	KorSciBert${}_{WA}$	89.79 (−0.03)	42.53 (−0.38)	-
Science and Tech	KorSciElectra	90.03	37.78	-
	KorSciElectra${}_{A}$	89.20 (−0.83)	35.78 (−2.00)	-
	KorSciElectra${}_{WA}$	90.04 (+0.01)	38.19 (+0.41)	-

**Table 4 sensors-23-03064-t004:** Examples.

AIDA-SC	AIDA-FC	KLUE-TC
[A, N, O, V, A] → [ANOVA]	[The, ore, tical] → [Theoretical]	[우리, 카드] → [우리카드] ^1^
[SP, S, S] → [SPSS]	[D, ata, se, ts] → [Data, set, s]	[카카오, 페이지] → [카카오페이지] ^2^
[영유, 아보, 육, 법] → [영유아, 보육, 법] ^3^	[Cons, tr, uc, ti, n, g] → [Construc, ting]	[에르, 난, 데스, 에, 게] → [에르난데스, 에게] ^4^
[UNK] → [χ2]	[UNK] → [햅틱 (Haptic)]	[UNK] → [챗봇 (Chatbot)]
[UNK] → [μg]	[UNK] → [펩타이드 (Peptide)]	[UNK] → [잰걸음 (Brisk walking)]
[UNK] → [카드뮴 (Cadmium)]	[UNK] → [에멀젼 (Emulsion)]	[UNK] → [셧다운 (Shutdown)]

^1^ 우리 (/Uri/) means “we”. 카드 (/Kadeu/) means “card”. 우리카드 (/Uri Kadeu/) represents the name of one of South Korea’s credit card companies. ^2^ 카카오 (/Kakao/) usually means “a fruit, an ingradient of chocolate”. 페이지 (/Peiji/) means “page”. 카카오페이지 (/KakaoPeiji/) is a monetized content platform. ^3^ 영유 (/Yeongyu/) is usually used in the news domain as an abbreviation for English kindergarten. 아보 (/Abo/) and 육 (/Yuk/) by itself has no special meaning. 법 (/Beop/) is “Law”. 영유아보육법 (/Yeongyua Boyuk Beop/) is a law for child care. ^4^ 에르 (/Ereu/), 난 (/Nan/), 데스 (/Deseu/), 에 (/e/), 게 (/ge/) by itself has no special meaning. 에르난데스 (/Ereunandeseu/) is “Hernández”. 에게 (/Ege/) means “for”.

## Data Availability

Not applicable.

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
