# Peer review of "Domain Word Extension Using Curriculum Learning"

_sensors, 2023, doi:10.3390/s23063064_

Round 1
Reviewer 1 Report
How are domain-related extended vocabularies determined? A clear acquisition process should be given in the manuscript. Is the example given in Figure 3 reasonable? Datasets should be a common term in many fields. Why should it be used as an extended term? The authors stated that the word "datasets" has a specific meaning different from previous. How is this determined? Is there a way to quickly determine the specific vocabulary of a specific domain, instead of being determined by domain experts based on domain knowledge?
How to complete embedding learning of extension words based on domain knowledge? The detailed operation process is not given in the manuscript. How to complete the integration of basic vocabulary embedding and extended vocabulary embedding? No detailed explanation is given in the manuscript.
Two problems in the current pre-training language model are given in the Abstract. There is a lack of in-depth discussion on these two problems and the contribution of the methods proposed in this paper to the problem solving.
These problems are the core innovation of this paper. However, the manuscript lacks a detailed discussion for these problems.
Reviewer 2 Report
Foreign characters should be changed to English, such as figure and table numbering.
Reviewer 3 Report
The paper's authors present a methodology for extending the vocabulary of pre-trained language models without retraining the model from scratch. The article is well-written and easy to follow. The topic might interest a broad audience of researchers working with transformer-based models for language processing.
The motivation behind the presented methodology is well justified. However, some potentially unclear steps in the procedure might make it challenging to reproduce similar results.
1. How do you decide how to extend the vocabulary? Is it done manually or automatically? In the case of words replaced with [UNK] token, it is a straightforward operation — and the replaced sequence to the vocabulary. More unclear is how to recognize and handle over-tokenized words like "datasets". Why you used the split into "data ##set #s" over "dataset ##s"? The original model does not know the words "data" and "set" as it tokenized the word into "d ##ata #se #ts, so it will not benefit from averaging embeddings for "data" and "set". "data" will be a combination of token "d" and "ata".
2. How many tokens were added to each model listed in Table 2?
3. In Table 2, for KoElextre(WA) and AIDA-SC — +0.25 != 89.21 - 89.50
There is a minor issue: several non-English words are in the article, mostly instead of "Table" and in the header.
Round 2
Reviewer 1 Report
In the current version, the author has solved all my previous questions. I suggest that the current version of the manuscript could be accepted.